# KI and WU Polyomavirus in Respiratory Samples of SARS-CoV-2 Infected Patients

**DOI:** 10.3390/microorganisms9061259

**Published:** 2021-06-09

**Authors:** Carla Prezioso, Ugo Moens, Giuseppe Oliveto, Gabriele Brazzini, Francesca Piacentini, Federica Frasca, Agnese Viscido, Mirko Scordio, Giuliana Guerrizio, Donatella Maria Rodio, Alessandra Pierangeli, Gabriella d’Ettorre, Ombretta Turriziani, Guido Antonelli, Carolina Scagnolari, Valeria Pietropaolo

**Affiliations:** 1Department of Public Health and Infectious Diseases, “Sapienza” University of Rome, 00185 Rome, Italy; carla.prezioso@uniroma1.it (C.P.); gabriele.brazzini@uniroma1.it (G.B.); piacentini.1854105@studenti.uniroma1.it (F.P.); gabriella.dettorre@uniroma1.it (G.d.); 2IRCSS San Raffaele Pisana, Microbiology of Chronic Neuro-Degenerative Pathologies, 00163 Rome, Italy; 3Department of Medical Biology, Faculty of Health Sciences, University of Tromsø—The Arctic University of Norway, 9037 Tromsø, Norway; ugo.moens@uit.no; 4Laboratory of Microbiology and Virology, Department of Molecular Medicine, “Sapienza” University of Rome, 00161 Rome, Italy; giuseppe.oliveto@uniroma1.it (G.O.); federica.frasca@uniroma1.it (F.F.); agnese.viscido@uniroma1.it (A.V.); mirko.scordio@uniroma1.it (M.S.); giuliana.guerrizio@uniroma1.it (G.G.); dona.rodio@gmail.com (D.M.R.); alessandra.pierangeli@uniroma1.it (A.P.); ombretta.turriziani@uniroma1.it (O.T.); guido.antonelli@uniroma1.it (G.A.); carolina.scagnolari@uniroma1.it (C.S.); 5Microbiology and Virology Unit, “Sapienza” University Hospital “Policlinico Umberto I”, 00161 Rome, Italy; 6Istituto Pasteur Italia, 00161 Rome, Italy

**Keywords:** Karolinska Institutet polyomavirus, Washington University polyomavirus, SARS-CoV-2, co-infection, oropharyngeal swab, NCCR sequencing

## Abstract

Severe Acute Respiratory Syndrome Coronavirus-2 (SARS-CoV-2) has been declared a global pandemic. Our goal was to determine whether co-infections with respiratory polyomaviruses, such as Karolinska Institutet polyomavirus (KIPyV) and Washington University polyomavirus (WUPyV) occur in SARS-CoV-2 infected patients. Oropharyngeal swabs from 150 individuals, 112 symptomatic COVID-19 patients and 38 healthcare workers not infected by SARS-CoV-2, were collected from March 2020 through May 2020 and tested for KIPyV and WUPyV DNA presence. Of the 112 SARS-CoV-2 positive patients, 27 (24.1%) were co-infected with KIPyV, 5 (4.5%) were positive for WUPyV, and 3 (2.7%) were infected simultaneously by KIPyV and WUPyV. Neither KIPyV nor WUPyV DNA was detected in samples of healthcare workers. Significant correlations were found in patients co-infected with SARS-CoV-2 and KIPyV (*p* < 0.05) and between SARS-CoV-2 cycle threshold values and KIPyV, WUPyV and KIPyV and WUPyV concurrently detected (*p* < 0.05). These results suggest that KIPyV and WUPyV may behave as opportunistic respiratory pathogens. Additional investigations are needed to understand the epidemiology and the prevalence of respiratory polyomavirus in COVID-19 patients and whether KIPyV and WUPyV could potentially drive viral interference or influence disease outcomes by upregulating SARS-CoV-2 replicative potential.

## 1. Introduction

In late December 2019, several cases of pneumonia of unknown origin were reported in Wuhan, China, and in early January 2020, a novel coronavirus denominated severe acute respiratory syndrome coronavirus 2 (SARS-CoV-2) was identified and defined as the etiological agent of coronavirus disease 2019 (COVID-19) [1,2]. Despite efforts to contain the disease in China, the virus spread globally and in March 2020 COVID-19 was declared a pandemic by the World Health Organization (WHO) [3].

SARS-CoV-2 is able to cause a wide range of clinical manifestations varying from asymptomatic or mild symptoms to severe illness and death [1,2,4]. Moreover, even though viral co-infections are able to influence the resultant disease pattern in the host, very few studies have looked at the disease outcomes in patients infected with SARS-CoV-2 and other pulmonary pathogens [5].

Co-infection with several viruses has been documented in patients with severe acute respiratory syndrome (SARS) and Middle East respiratory syndrome (MERS). Among patients infected by SARS-CoV-2, the knowledge remains limited. In this regard, it has been established that the prevalence of co-infection with respiratory pathogens was variable among COVID-19 patients and could be up to 50% among non-survivors [6]. However, to date, there is no data on the relationship between human polyomavirus (HPyVs)/SARS-CoV-2 co-infection and clinical and epidemiological profile of COVID-19 patients.

Karolinska Institutet Polyomavirus (KIPyV) and Washington University Polyomavirus (WUPyV) were discovered in 2007 in nasopharyngeal samples from patients with respiratory symptoms [7,8]. So far, whether KIPyV and WUPyV are genuine respiratory pathogens or opportunistic co-infectors has not been established, although KIPyV transcripts were identified without other viral sequences in cases of otherwise healthy individuals with severe respiratory symptoms and WUPyV sequences were found in WUPyV-associated bronchitis [9,10,11].

Both KIPyV and WUPyV genomes, like that of the other HPyVs, consist of the early region, the late region and of the noncoding control region (NCCR) [12]. The early region codes for regulatory proteins involved in replication and transcription of the viral genome. The major early proteins are large T- and small t-antigen (sT). The late region codes for the structural proteins of which VP1 is the major capsid protein, while VP2 and VP3 are the minor capsid proteins [13,14]. Interposed between the early and late region there is the NCCR, a sequence that does not code for viral proteins. While the protein coding regions reveal a strong sequence conservation, the NCCR could exhibit variation consisting in deletions, duplications and rearrangements. Genetic variability in the NCCR of BK (BKPyV) and JC polyomavirus (JCPyV) may affect viral replication and determine clinical consequences [12]. Whether mutations in KIPyV and WUPyV NCCRs may have an effect on their pathogenic properties remain to be determined.

Since to date a picture of the prevalence of KIPyV and WUPyV in the occurrence of SARS-CoV-2 infection during the early outbreak period of COVID-19 has not emerged, our goal was to determine whether co-infections with KIPyV and WUPyV occur in a significant subset of SARS-CoV-2 infected patients. Because a co-infection with other pulmonary pathogens can carry significant risks on the clinical outcomes of COVID-19 patients representing an important concern for clinicians, understanding the epidemiology and prevalence of HPyVs in these patients will help to better document the SARS-CoV-2 co-infection.

## 2. Materials and Methods

### 2.1. Study Population and Sample Collection

Oropharyngeal swabs collected from 150 individuals, 112 symptomatic COVID-19 patients and 38 healthcare workers not infected by SARS-CoV-2, were tested for KIPyV and WUPyV DNA presence. Diagnosis of SARS-CoV-2 was previously performed by real-time RT-PCR assay (RealStar SARS-CoV2 RT-PCR, Altona Diagnostics) [15] from March 2020 to May 2020 at Policlinico Umberto I Hospital, Rome, Italy. Viruses identified in each specimen were documented and summarized as co-infection, single infection or negatives. Specifically, co-infection was defined as the presence of SARS-CoV-2 with at least one HPyV or presence of both HPyVs in the same specimen. A single infection was considered when only SARS-CoV-2 or HPyVs was detected and, finally, a negative result was considered as no detection of SARS-CoV-2 and/or HPyVs. Demographic data are reported in Table 1. This study was approved by the institutional review board (Policlinico Umberto I Hospital, Sapienza, University of Rome, Italy) and the Ethics Committee (Sapienza, University of Rome, Italy).

### 2.2. Virological Analysis

Testing for the presence of KIPyV and WUPyV was performed using three hundred microliters of oropharyngeal samples subjected firstly to DNA extraction using DNeasy^®^ Blood and Tissue Kit (QIAGEN, S.p.A, Milan, Italy) according to the manufacturer’s instructions. To test DNA quality, β-globin PCR was carried out as previously described [16]. All β-globin-positive specimens were tested for KIPyV and WUPyV DNA by real-time PCR as detailed previously with primers and probes targeting the VP2-3 region of KIPyV and VP region of WUPyV [17]. Plasmids containing the KIPyV (pcDKIER (#37094)) and WUPyV (pcDWUER (#37093)) genomes and distilled water were used as a positive and negative control, respectively. All samples were tested in triplicate and standard precautions were taken to prevent contamination during amplification procedures.

### 2.3. Amplification, Sequencing and Analysis of KIPyV and WUPyV NCCRs

KIPyV and WUPyV DNA-positive samples were subsequently amplified for NCCR region following published protocols [18]. By electrophoresis in 2% agarose gel, PCR products were analyzed, stained with ethidium bromide and observed under UV light. The amplified products were purified using the MinElute PCR Purification Kit (QIAGEN, Milan, Italy) and sequenced in a dedicated facility (Bio-Fab research s.r.l., Rome, Italy). The obtained sequences were compared to reference sequences deposited in GenBank. Sequence alignments were performed with ClustalW2 at the European Molecular Biology Laboratory–European Bioinformatics Institute (EMBL-EBI) website using default parameters [19].

### 2.4. Statistical Analysis

The continuous variables were expressed both as mean ±SD and as median and range. All studied features were analyzed with non-parametric tests, like Chi-square (χ^2^) test, Kruskal-Wallis test and Mann-Whitney U for unmatched data. Statistical analyses were performed with SPSS v.25.0 for Windows: a p value less than 0.05 was considered statistically significant.

## 3. Results and Discussion

In order to improve the understanding on the epidemiological features of HPyVs in patients affected by COVID-19, 112 oropharyngeal swabs from hospitalized patients with symptomatic COVID-19 and 38 oropharyngeal swabs from healthcare workers not infected by SARS-CoV-2, for a total of 150 samples, were tested for the presence of KIPyV and WUPyV DNA. Overall, KIPyV DNA was detected in 27/150 (18%) swabs, WUPyV DNA in 5/150 samples (3.33%) and DNAs of both KI and WU viruses were found in 3/150 (2%) oropharyngeal specimens (Table 2).

Specifically, of the 112 SARS-CoV-2 positive patients, 27 (24.1%) were co-infected with KIPyV, in 5 (4.5%) SARS-CoV-2/WUPyV co-infection was found and 3 (2.7%) were infected simultaneously by KIPyV and WUPyV (Table 2).

Regarding the group of 38 healthcare workers, neither KIPyV nor WUPyV DNA was detected in oropharyngeal swab samples. Although the healthcare workers sample number was low and further investigation with larger cohort is required to accurately estimate the distribution of WUPyV and KIPyV infections, our results are consistent with previous reports indicating that respiratory HPyVs were commonly detected both in immunocompromised children and adults, in immunocompetent children and rarely detected in immunocompetent adults [20,21,22,23].

Interestingly, the statistical analysis evidenced a significant difference in the rate of KIPyV infection between patients co-infected by SARS-CoV-2 and the SARS-CoV-2 negative group (*p* < 0.05, OR > 1) (Table 2).

As reported in the literature, since KIPyV is often found in other various sample types such as stool, blood, urine, lymph node, spleen, cerebrospinal fluid, lung, tonsil and adenoids, a possible explanation about the significant co-infection SARS-CoV-2-KIPyV could be that KIPyV represents an opportunistic co-infector rather than a genuine respiratory pathogen [18,24,25].

Moreover, our results could suggest that, although HPyVs infections occur early in childhood and persist throughout life enhancing host heterologous immunity, as observed in persistently infected individuals who experience constant, low-level of antigenic stimulation resulting in protective cross-immunity via different processes (innate immune stimulation, activation of CD4^+^ or CD8^+^ T cells, cross-reactive CD8^+^ T cells) [26,27], SARS-CoV-2 infection may significantly alter the host’s immune response and allow a KIPyV and, in part also WUPyV, surge in replication.

No statistically significant difference was found in KIPyV and WUPyV prevalence according to age and gender, although patients tested positive for SARS-CoV-2 and co-infected with KI and WU viruses were predominantly females, similar to findings describing characteristics of SARS-CoV-2 infection in Italy in which cases were more common among women (53.1%), mainly among the elderly population [4].

The role of viral load in respiratory viral infection is highly debated. It has been proposed that the viral load of some respiratory viruses correlates with disease severity [28] and recently also SARS-CoV-2 high viral load was associated with COVID-19 [29]. Considering viral co-infections, human coronavirus NL63 RNA levels in patients with a co-infection are often much lower than the load in the mono-infected patients [29]. In this study, the evaluation of SARS-CoV-2 cycle threshold (Ct) values in relation with the presence of KIPyV and WUPyV DNA showed that Ct values were lower (corresponding to a higher viral RNA concentration) in SARS-CoV-2 patients co-infected with KIPyV and WUPyV simultaneously (Ct median value 18.00 (range: 12.88–22.00)) with respect to all other groups. Moreover, Ct values were lower (21.66 (12.88–25.75)) among SARS-CoV-2–WUPyV-infected patients compared with SARS-CoV-2-KIPyV-infected patients (median Ct value: 25.63 (12.88–38.62)) (Table 3).

However, statistical analysis revealed that SARS-CoV-2 Ct values were significantly correlated with the detection of KIPyV, WUPyV and KIPyV and WUPyV co-infection (*p* < 0.05), confirming that viruses do not competitively suppress the replication of another co-infecting virus and further supporting the hypothesis of KIPyV and WUPyV as opportunistic co-infectors (Figure 1).

Since little is known on the genetic diversity of KIPyV and WUPyV NCCRs and on the biological relevance in terms of viral transcription, replication and possible pathogenic properties, the genetic analysis of NCCRs, obtained from 27 out of 112 (24.11%) positive KIPyV samples, was performed. Alignments of 27 isolates from oropharyngeal swabs from hospitalized patients co-infected by KIPyV and SARS-CoV-2 showed a NCCR characterized by a high degree of homology compared with Stockholm 60 reference sequence (Genbank accession number NC_009238) [7]. This result supports previous observations that larger KIPyV NCCR rearrangements, as seen for the NCCRs of clinical BKPyV and JCPyV isolates, seem to be rare [12]. The amplified NCCRs obtained from five WUPyV samples were similar to the strain deposited in GenBank under the accession number EF444549 [8] since, in one sample, the G54A and T59G mutations were observed. Both G54A and T59G are most common point mutations and, as in this study, often present simultaneously [12]. Because most variations described for WUPyV are single or few point mutations, they may not destroy or create novel binding sites and may not have an effect on WUPyV promoter activity and viral replication.

In conclusion, although to the best of our knowledge this is the first study on HPyVs in the contest of SARS-CoV-2 infection, additional investigations are needed to understand the epidemiology and the prevalence of respiratory polyomavirus in COVID-19 patients and whether KIPyV and WUPyV could potentially drive viral interference or influence disease outcomes by upregulating SARS-CoV-2 replicative potential.

In fact, although the incidence in respiratory samples of SARS-CoV-2 with other respiratory viruses was examined by several groups, the impact on the clinical outcome is less known. In SARS-CoV-2 positive patients, the rates of co-infection with other viruses were found to be lower, no significant differences were established, and clinical implications were not determined or were not conclusive [30,31,32]. An in vitro study demonstrated that infection of different cell lines, even cells that normally do not support SARS-CoV2 infection, with influenza A virus increased their susceptibility to SARS-CoV-2 infection [33]. In a mice model, influenza A virus and SARS-CoV-2 co-infection resulted in increased SARS-CoV-2 viral load and more severe lung damage [33]. It is possible to speculate that infection with influenza A virus induces expression of the SARS-CoV-2 receptor ACE2, which can (partially) explain the increased susceptibility for SARS-CoV-2 [33].

The effect of co-infection with KIPyV or WUPyV on SARS-CoV-2 replication and clinical outcome has not been investigated. The lack of suitable cell culture systems and animal models has limited research on the infectivity and pathogenicity of these HPyVs. However, a recent study showed that WUPyV can be propagated in primary human airway epithelial cells, but it was not tested whether these cells are permissive for KIPyV [34]. Interestingly, these cells were also used to isolate SARS-CoV-2 that caused the outbreak in Wuhan, China [1], which would allow WUPyV and SARS-CoV-2 co-infection experiments.

Further research is required to unveil whether KIPyV or WUPyV can increase the sensitivity for SARS-CoV-2 infection and whether coinfection may exacerbate the pathogenesis of this virus.

## Figures and Tables

**Figure 1 microorganisms-09-01259-f001:**
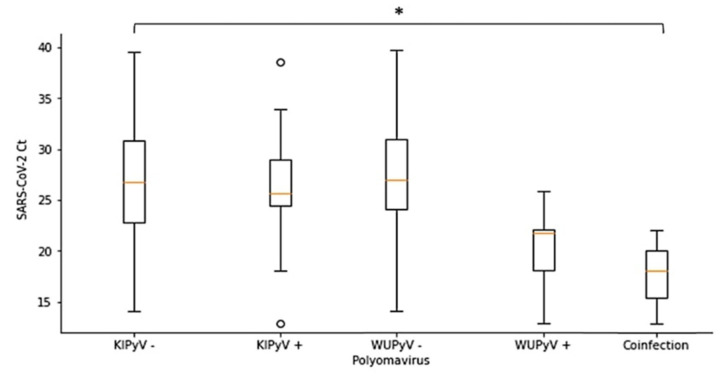
Distribution of Ct of SARS-CoV-2 according to the presence or absence of KIPyV and WUPyV. *: *p* < 0.05 (Kruskal-Wallis test), o: Outliers.

**Table 1 microorganisms-09-01259-t001:** Demographic characteristics.

FEATURES		POPULATION	
**Enrolled Population,** *n*		150	
Sex, *n* (M/F)		**M**	**F**
		79/150 (52.7%)	71/150 (47.3%)
Mean age, years (SD)		59.06 (±16.52)	
Median age, years (Range)		60.71 (92.84–22.1)	
**SARS-CoV-2 patients**	**Enrolled population,** *n*	112	
	Sex, *n* (M/F)	**M**	**F**
		51/112 (45.5%)	61/112 (54.5%)
	Mean age, years (SD)	61.42 (±16.22)	
	Median age, years (Range)	62.45 (22.1–92.84)	
**Healthcare workers**	**Enrolled population,** *n*	38	
	Sex, *n* (M/F)	**M**	**F**
		18/38 (47.4%)	20/38 (52.6%)
	Mean age, years (SD)	52.11 (±15.42)	
	Median age, years (Range)	49.35 (23.94–89.0)	

SD: Standard Deviation, M: male, F: female.

**Table 2 microorganisms-09-01259-t002:** Analysis of KIPyV and WUPyV prevalence in the examined population.

Polyomavirus		Total Population	SARS-CoV-2 +	SARS-CoV-2 −	χ^2^	OR
KIPyV, n	**P**	27/150 (18%)	27/112 (24.11%)	0/38 (0%)	***p* < 0.05**	>1
	**N**	123/150 (82%)	85/112 (75.89%)	38/38 (100%)		
WUPyV, n	**P**	5/150 (3.33%)	5/112 (4.46%)	0/38 (0%)	*p* > 0.05	>1
	**N**	145/150 (96.67%)	107/112 (95.54%)	38/38 (100%)		
KIPyV-WUPyV co-infection, n		3/150 (2%)	3/112 (2.68%)	0/38 (0%)	*p* > 0.05	>1

OR: Odds Ratio; P: positive; N: negative.

**Table 3 microorganisms-09-01259-t003:** Analysis of SARS-CoV-2 Ct values according to the KIPyV and WUPyV detection.

Polyomavirus		Ct Median Values (Range)
KIPyV	P	25.63 (12.88–38.62)
	N	26.75 (14.02–39.53)
WUPyV	P	21.66 (12.88–25.75)
	N	26.86 (14.02–39.53)
KIPyV-WUPyV co-infection		18.00 (12.88–22.00)

P: positive; N: negative.

## Data Availability

Data is contained within the article.

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
