# Peer review of "KI and WU Polyomavirus in Respiratory Samples of SARS-CoV-2 Infected Patients"

_microorganisms, 2021, doi:10.3390/microorganisms9061259_

Round 1
Reviewer 1 Report
In this manuscript, Prezioso et al. suggest that the infection by KI and/or WU polyomaviruses may potentially be opportunistic during SARS-CoV-2 one. Their hypothesis is based on direct clear-cut detection approach that is used routinely and established as a fast standard procedure for viruses’ detection. The paper is clearly written and the data are described well enough. However, one major caveat in Prezioso et al.’s investigation is the absence of any information/data about the type of medicine/support (if any) their Covid-19 patients were receiving for treatment. This is important to fully consider by our time because contemporary medicines do systematically modulate the immunity of patients during their prescribed treatment(s). To this end, Prezioso et al. require necessarily to address this point clearly and rule out any link between the treatment Covid-19 patients received (if any) and the prevalence of KI and/or WU polyomaviruses infection.
Reviewer 2 Report
This is an iteresting paper reporting the frequancy of the association among SARS-CoV-2 infection and the two respiratory human polyomaviridae KIPyV and WUPyV.
The work was conducted with scientific precision, methods are appropriate and results were clearly shown and discussed.
Main strenghts are the novelty and the originality of the study design; main weaknesses are the limited number of the cohort of patient included in the study and the clinical relevance of the results
My main concern is the translational significance of the study, that means the co-infection with respiratory poliomaviridae can influence the clinical course of the SARS-CoV-2 illness? The clinical findings of the two group of the patients (with and without polyomaviridae co-infection) are somehow different?
I think these information have to be added.
Line 54: a reference should be added in the text.
Round 2
Reviewer 1 Report
The manuscript by Prezioso et al. is now enclosed and should be accepted for publication.